# Therapy Strategies for Children Suffering from Inflammatory Bowel Disease (IBD)—A Narrative Review

**DOI:** 10.3390/children9050617

**Published:** 2022-04-26

**Authors:** Corinne Légeret, Raoul Furlano, Henrik Köhler

**Affiliations:** 1University Children’s Hospital of Basel, 4056 Basel, Switzerland; raoul.furlano@ukbb.ch; 2Children’s Hospital Kantonsspital Aarau, 5000 Aarau, Switzerland; henrik.koehler@ksa.ch

**Keywords:** children, inflammatory bowel disease, treatment, guidelines

## Abstract

The incidence of inflammatory bowel disease (IBD) is increasing, and more children at a younger age are affected. The pathogenesis seems to be an interaction of microbial factors, a sensitivity of the immune system, and the intestinal barrier, leading to an inappropriate immune response. Not only has the role of biological agents become more important in the last decade in the treatment of children and adolescents, but also new insights into the composition of the gastrointestinal microbiome and personal diet implications have increased our understanding of the disease and opened up potential therapeutic pathways. This narrative review provides an overview of current recommendations, therapeutic options, drug monitoring, and practical guidelines for paediatricians involved with paediatric IBD patients. Furthermore, the off-label use of potential helpful drugs in the treatment of these patients is discussed.

## 1. Introduction

Inflammatory bowel disease (IBD) is a blanket term for a chronic inflammation affecting the gastrointestinal tract, which can be classified into ulcerative colitis (UC), Crohn’s disease (CD), and indeterminate colitis (IC). UC is a continuous, mucosal inflammation of the rectocolon and can, especially in children, present with atypical features (rectal sparing, backwash ileitis [1]). CD, in contrast, is defined by a transmural, non-continuous inflammation, affecting the gastrointestinal tract potentially anywhere from the mouth to the anus. In children, the inflammation is mainly found in the ileocolonic region [2]. Indeterminate colitis is characterized by typically clinical and endoscopic signs of a chronic inflammation of the colon without having specific features for UC or CD but rather subtle features of both. In children, IC is twice as common as in adults, but in 80% of those cases, patients will be classified as having CD or UC once they are adults [3].

The natural history and clinical presentation of IBD are different from those in adult patients, but paediatric gastroenterologists are confronted with a rapidly growing therapeutic armoury, which has primarily been used in adults. This review aims to examine current therapeutic possibilities and its evidence regarding the response.

## 2. Epidemiology

Worldwide, the prevalence of IBD is steadily increasing. Recent data indicate higher rates of paediatric CD compared to UC in Europe and North America, except for Scandinavia and Southern and Eastern Europe [4]. In 2018, Sykora et al. [5] performed a systematic review of 140 studies from 38 different countries: they found the highest annual paediatric incidence of IBD with 23/100,000 in Europe, 15.2/100,000 in North America, and 11.4/100,000 in Asia/middle East. Whereas the incidence of UC has remained stable, the global incidence rates of IBD seem to have risen mainly due to an increase in paediatric CD and IC.

IBD is thought to have a strong genetic component, as a positive family history represents a high risk to develop the disease at all ages; affected patients often present at a younger age and are more likely to experience extra-intestinal manifestations. Large genome-wide association studies of IBD found more than 200 common loci associated with IBD [6], which account for less than 30% of the variation in the disease. This is in contrast with children, who are diagnosed with very early-onset IBD (VEO-IBD; diagnosis is made before the age of 6 years), where in 15–20%, a monogenic aetiology can be detected [7]. Monogenic variants of VEO-IBD can be classified into five distinct groups (epithelial barrier defects; phagocytic defects; T and B cell defects; T regulatory cells and signalling; hyper- and auto-inflammatory conditions [8]) and should be treated as its own entity (see Table 1).

As in IBD patients, genes per se seem to play a minor role, environment interaction must be considered in the pathogenesis. This hypothesis is underlined by the fact that IBD is increased among immigrants to a country with a high prevalence of IBD [9], which seems to be associated with a ‘westernization’ of the gut microbiome, caused by diets high in animal fats and sugar, whilst being low in fruits and vegetables—all known risk factors for IBD, as it may change the microbiome [10].

## 3. Treatment Goal

The overall goal for clinicians in treating children suffering from IBD is not only to achieve clinical remission but also to attain mucosal healing in order to assure a normal growth, a good quality of life, and the prevention of hospitalization and surgery. Especially in children, repetitive performance of endoscopies is not feasible for monitoring the course of the disease, as children need to undergo a bowel cleanout and an anaesthetic, both potentially traumatizing experiences; therefore, laboratory parameters and clinical disease activity indexes are used to assess the level of inflammation: the PUCAI (Paediatric Ulcerative Colitis Activity Index) is based on six questions, can differentiate accurately activity states, and is used throughout the world since its establishment in 2007 [11]. The equivalent tool to assess disease activity and response to treatment in clinical practice of paediatric CD is the Paediatric Crohn’s Disease Activity Index (PCDAI) [12]. In most trials and statistical analysis, the above-mentioned scores are used to compare the effect of medication, which, in general, show a good correlation with endoscopic findings in children. Specifically, paediatric IBD activity scores are more reliable in children with a severe disease than the ones having a mild or moderate course [13,14]. Laboratory parameters, which are commonly used to assess the activity of the disease, are faecal Calprotectin, Albumin, Erythrocyte Sedimentation Rate (ESR), and C-Reactive protein (CRP).

Although children and adolescents present with a more severe phenotype and a longer disease duration, paradoxically, there is a long delay before paediatric trials of new drugs are started: the first antitumour necrosis factor (TNF) inhibitor, Infliximab (IFX), has been registered for use in adult CD in 1999, the first prospective trial in children was only performed in 2007 [15], and a similar delay occurred for Adalimumab. Differences between children and adults exist not only in disease presentation, but also in regard to the absorption, distribution, and metabolism of drugs. This illustrates the need for controlled, randomized paediatric trials and recommendations based on those findings. Due to unknown long-term complications, newly developed drugs are firstly used in adults for ethical reasons; therefore, most paediatric guidelines rely on results from large adult trials, underlined by findings from small studies in children. Most trials or guidelines focus on one subgroup of IBD; we hereby summarize and discuss current guidelines to provide an overview on the advantages and disadvantages of currently used drugs in IBD. We performed a PubMed search for literature, which has been published within the last 5 years (January 2017–January 2022) using the terms ‘paediatric IBD’, ‘treatment of paediatric CD/UC’, ‘guidelines in treating children with UC/CD’, and ‘biologics/TNF-alpha/adalimumab/golimumab/JAK-inhibitor in children with IBD’, mainly focusing on randomized controlled trials, but also included main adult trials, on which paediatric recommendations rely.

### Treatment Approaches

Management principles are used not only to induce remission, but also to maintain it. Some medications can be used to induce remission, and others to maintain it, and sometimes they can be used for both phases of the disease. In paediatrics, a step-up treatment is applied, meaning that initially less-effective drugs with a mild risk of adverse effects are applied, with the possibility to escalate the treatment to more potent medications with possibly more side effects. Below, the reader will be given an overview over different medication classes in a ‘step-up’ order with practical recommendations.

## 4. Pharmakotherapy

### 4.1. 5-ASA

5-aminosalicylic acid (5-ASA) has several anti-inflammatory effects, acts locally in the intestinal mucosa, and can be administered orally or rectally. To inhibit the absorption and inactivation of 5-ASA in the small intestine, coated and time-dependent preparations have been developed, providing release of it throughout the gastrointestinal tract. A meta-analysis showed that mucosal healing in adults suffering from UC was achieved by 37% of almost 4000 patients using oral 5-ASA and by half of over 2500 patients using rectally applied 5-ASA [16]. Clinical paediatric trials with 5-ASA typically involved only few patients but did confirm a benefit for patients with UC [17,18]. Due to a mild side-effect profile and good efficacy, 5-ASA is recommended as a first-line induction therapy for mild or moderately active UC and for maintenance of remission in children [19]. It is still often prescribed for maintenance in CD; however, data regarding the effect are inconsistent, and it is therefore not mentioned in the guidelines for paediatric CD [20].

### 4.2. Corticosteroids

For all paediatric patients suffering from IBD, the only indication to start treatment with corticosteroids is to induce remission but not to maintain it. UC patients who either have not achieved remission despite a proper treatment with 5-ASA or have moderate to severe colitis are recommended to start on oral steroids; those with severe symptoms might benefit from intravenous corticosteroid therapy [19]. In children with luminal CD, when exclusive enteral nutrition (EEN) cannot be applied, steroids should be prescribed for inducing remission. In cases of mild ileocaecal disease, budesonide is preferable to prednisolone [20] due to its potentially less adverse events and an ileal release. Oral prednisone is recommended for initiation at a dose of 1 mg/kg once daily (max. daily dose of 40–60 mg) for 2–4 weeks and should then be tapered over the course of 8–10 weeks [19,21].

### 4.3. Immunomodulators

Azathioprine and mercaptopurine—immunomodulators that may require up to 14 weeks to have maximum effectiveness—will be discussed. It is recommended for all IBD patients who have already reached remission to maintain it. A Cochrane review of 881 adult patients [22] showed that azathioprine, compared to placebo, is significantly more effective in maintaining remission without using steroids; available data for children are weaker. Before starting the treatment, available data encourage the determination of the thiopurine-methyltransferase (TPMT) geno- or phenotype to identify patients at higher risk for an early and severe myelosuppression [23]. In patients failing the treatment, it is helpful to measure the metabolites (6-methylmercaptopurine (6-MMP) and 6-thioguanine (6-TGN)) to assess whether the levels are within the therapeutic range and/or to assess compliance of drug use (see Table 2). The risk for IBD patients receiving thiopurines to develop lymphomas and non-melanoma skin cancers is small but cannot be neglected: in the largest adult trial, which included over 19,000 adult patients, a multivariate-adjusted hazard ratio of 5.3 was calculated between patients receiving thiopurines and thiopurine-naïve patients [24]. A meta-analysis [25] of eighteen studies found an overall standardized incidence ratio of 4.9 for lymphoma. Risk factors associated with the development of hepatosplenic T-cell lymphoma include at least two years of exposure to thiopurine, with male gender and the age under 35 years. As a primary infection with the Epstein–Barr virus during the treatment with thiopurine seems to be associated with an increased risk for lymphomas [17] as well, it is recommended to perform correspondent serologies before starting the treatment. It is of importance to notice that the increased risk normalizes after the discontinuation of therapy.

Data regarding the effectiveness of methotrexate (MTX) in adult UC patients for induction or maintenance are weak, as it is for children; therefore, it is recommended to only consider MTX in paediatric UC patients who fail to respond to thiopurines or if alternatives are not available [19]. This is different for paediatric CD patients, where a systematic review showed that 37% of patients were in clinical remission after 12 months [26]. Therefore, MTX became a first-choice immunomodulator to keep children with CD in remission [20].

### 4.4. Anti-TNF-α

Infliximab (IFX) is a monoclonal antibody against tumour necrosis factor-alpha (TNF-α) and was the first biological, which was licensed for children with IBD in 2006. The first large, prospective, multicentre, randomized controlled trial (RCT, REACH study) included 112 children with moderate or severe active CD and confirmed IFX to be effective in both the induction and maintenance of remission [27]. Children with a risk factor for a poor course in CD (extensive disease, deep colonic ulcers, perianal or stricturing disease [20]) and in UC (not enough clinical response to 5-ASA and immunomodulators, chronically active UC, or steroid-dependent [19]), are recommended to be put on IFX to induce and maintain remission. In a recent multicentre RCT, treatment-naïve paediatric patients newly diagnosed with CD were assigned to different groups, which either received IFX, EEN, or prednisolone. Clinical and endoscopic assessment ten weeks into treatment, showed that a significantly higher proportion of participants on IFX were in remission, compared to the group that received conventional treatment (EEN/prednisolone) [28]. Other studies [29,30] reported similar results in using IFX as a first-line treatment to induce remission, especially in more severe presentations of IBD.

In young children, clearance of IFX is more rapid; therefore, higher doses and shorter intervals of 4–5 weeks may be needed [31]. The testing of drug level and antidrug antibodies has emerged to guide therapeutic decisions. Nowadays, the aspired drug level is higher than in the early days of implementing IFX in paediatrics. There are no clear guidelines, but concentrations greater than 7 μg/mL postinduction at week 14 and above 5 μg/mL during maintenance are associated with better chances of mucosal healing. For patients with an ongoing flare, IFX should not be stopped unless its concentration is greater than 10 μg/mL—this recommendation can be applied to all IBD subtypes. For patients suffering from perianal fistulizing disease, a drug level of approximately 12 μg/mL should be aimed for [32].

Before starting treatment with thiopurine drugs one may determine Thiopurine S-methyltransferase (TPMT) geno-/phenotype. TPMT is the key enzyme in metabolizing thiopurine drugs and TPMT heterozygote patients are at intermediate risk of bone marrow toxicity. Values for enzyme activity are not reliable if blood transfusion has been received within previous 10 weeks. However, even normal metabolizers need close monitoring of the blood count in the first weeks of therapy; therefore, the benefit of TMPT testing prior to thiopurine treatment is controversial.

In patients with normal metabolism, the recommended dose of azathioprine is 2–2.5 mg/kg or 1–1.5 mg/kg once daily. Metabolite profiles can be assessed as follows.

Albumin and inflammatory markers need not only to be determined to assess the level of inflammation, but also to decide about the dosage of IFX, as patients with a high inflammatory load need higher dosing of IFX to reach target levels [33]. It is of importance to know that higher drug levels do not correlate with an increased risk for infections or any other adverse events [34].

Proposed hypotheses explaining the loss of response to IFX include the formation of drug antibodies, a high inflammatory load of the disease, a rapid turnover of the drug, and the development of an alternate inflammatory pathway. Current guidelines state (see Figure 1) that, in children with a lost response to IFX, Adalimumab (ADA) should be considered for treatment [20] (Table 3), a totally humanized monoclonal biological agent belonging to the tumour necrosis factor blockers and approved for children since 2012. There are no paediatric trials that compare IFX directly to ADA. Looking at data from adults, it was shown that, in anti-TNF naïve patients who were treated with IFX or ADA, no significant difference was found in terms of the need for hospitalization due to disease exacerbation nor steroid-free clinical response after one and two years [35,36]. Based on such results, current recommendations suggest that, in paediatric anti-TNF naïve patients, IFX or ADA can be offered, considering the administration route, costs, availability, and patient preferences. In 2015, the patent for IFX expired, and a number of biosimilars (biological products that are very similar to the original product in regard to potency, safety, and purity) have obtained FDA approval. Trials in paediatric IBD patients showed the same clinical response, achieving remission in participants receiving an induction therapy with a biosimilar [37]. In a prospective study of 39 paediatric IBD patients in remission who were switched from IFX originator to Remsima^®^, no patient had a relapse, and no serious event occurred [38] during the 8 months of follow-up. Based on such findings, the current recommendations are to either start induction treatment with a biosimilar or only switch patients from originator to biosimilar once the patient is in clinical remission.

Immunogenicity is a well-known contributor to loss of response: anti-TNF are large proteins, which trigger the production of antibodies. In adding an immunomodulator to the anti-TNF treatment, it was hypothesized to reduce immunogenicity and therefore prevent loss of response. In adult biologic and immunomodulator naïve CD patients, the SONIC trial was the first randomized controlled trial showing that patients receiving a double treatment (IFX and immunomodulator) had higher drug and lower antibody levels and, overall, a better endoscopic outcome [39]. Similar results have been published from paediatric randomized trials, demonstrating benefits from combination therapies [40,41]. Therefore, European guidelines [20] recommend induction therapy with IFX in combination with an immunomodulator—a discontinuation of the immunomodulator after 6 months, especially in boys, may be considered when good drug levels are present.

Another fully human monoclonal anti-TNF-α drug is Golimumab, which was approved in 2013 for adult patients with moderate to severe active UC. It is not yet licensed for use in children, but small studies with 35 paediatric participants with UC showed 54% mucosal healing at week 6 with no significant clinical safety concern [42].

### 4.5. What Are the Options after a Failure of Anti-TNF-α?

A third of IBD patients need a change in treatment due to adverse events, allergic reactions, or primary or secondary loss of response [43]. Vedolizumab is a biological agent, being the most widely used in paediatrics: a humanized antagonist of integrin, characterized by a gut selectivity of action, which inhibits a migration of T-lymphocytes into inflamed intestinal tissue. Adult data proved its efficacy and safety in patients with moderate to severe UC and CD [43,44], but it has a slow induction rate of around 14 weeks and seems slightly less effective in CD patients, compared to the ones suffering from UC. The same results have been shown for children [45,46]. It is important to note that all performed paediatric trials are small, and no data are available for therapy-naïve children, as it still represents an off-label use in paediatrics.

Another biological drug that is currently being investigated for children with CD is Ustekinumab: a fully human immunoglobulin G1κ monoclonal antibody that binds to the P40 protein subunit of interleukins IL-12 and IL-23, which are involved in inflammatory response and T-cell differentiation. A small paediatric trial of 44 CD patients showed that 25% were in clinical remission at week 16 [47]. In this phase I study, the most frequent adverse event was exacerbation of the disease. A retrospective multicentre study affiliated to the IBD Interest and Porto group of 69 children with CD and previous TNF-α treatment showed that clinical remission was achieved by two-thirds of children at 3 months [48].

In adults, Tofacitinib (Janus kinase 1 and 3 inhibitor, downregulating several cytokines such as IL-6, -11, -4, -7, -9) has been approved for both the induction and maintenance of remission in patients with UC after efficacy was demonstrated in 2017: 40% of patients participating in a phase III randomized controlled trial achieved remission at week 52 [49]. In a phase IIb clinical trial with adult CD patients, clinical response was not significantly different compared to a treatment with placebo [50]. Janus kinase (JAK) inhibitors are small molecules with the clear advantage of oral administration, rapid onset of action, a half-life of 3 h, and lack of immunogenicity. JAK inhibitors have shown beneficial effects in patients with IBD. Tofacitinib is the first JAK inhibitor approved by FDA for adult patients with moderate-to-severe UC. A meta-analysis showed that mucosal healing was achieved in 48% of adult patients with UC at week 8 [49]. Results for Tofacitinib in patients with CD are contrary: patients in clinical studies failed to achieve remission. As JAKs play a key role in haematopoiesis, full blood counts should be periodically performed. Currently, there are a wide range of different subtype-selective JAK inhibitors under development, not only for the treatment of IBD, but also for other inflammatory conditions. In the CELEST trial, for example, in adult CD patients receiving Upadacitinib (selective to JAK1), clinical remission was achieved in 21% [51]. So far, only very small observational cohort studies in children with IBD receiving Tofacitinib are available: of 21 patients (UC and IBD-U), 43% showed a good clinical response at the end of the 12-week induction, and at 52 weeks, 33% were still on Tofacitinib and steroid-free, whereas the others showed a refractory disease [52].

## 5. Exclusive Enteral Nutrition/Diet

Liquid nutritional therapy has been found to be effective in achieving remission in CD since the 1970s. It is classified by the protein source and subdivided into elemental (amino acids), semi-elemental (hydrolyzed protein), and polymeric (whole-protein) diets, summarized under the name exclusive enteral nutrition (EEN). There is now a lot of literature supporting the use on nutritional therapy in CD, but the mechanism of action and ideal composition of formula are still unknown. A meta-analysis [53] that included over 1000 patients found no difference between elemental, semi-elemental, and non-elemental diets. 83% of EEN patients achieved remission, compared to 61% of patients receiving steroids. Patients receiving EEN were more likely to discontinue the treatment due to inability to tolerate it (unpalatable, intolerance of nasogastric tube, etc.) than the steroid groups. Based on a plethora of data, it is still recommended to bring children with luminal CD into remission [20] with EEN. Yu et al. [54] report a meta-analysis that did not show significant difference between EEN and steroids in inducing remission, but due to a milder profile of side effects, EEN is preferable. Unfortunately, it has been demonstrated that after an EEN-induced remission, faecal calprotectin rose in children within the first 2 to 3 weeks of food reintroduction [55]; therefore, many studies on different nutrient-based dietary therapies to maintain remission have been and still are being performed. Most data are available for maintenance enteral nutrition (MEN), meaning enteral nutrition formula, which contains a percentage of the total caloric intake. A meta-analysis of 392 paediatric CD patients using MEN as an additional therapy for immunosuppressant treatment showed benefits compared to a comparator group [56], but this was dependent on the amount of MEN: a daily consumption of over 35% of energy intake as MEN showed a significantly lower relapse rate compared to the control groups in 80% of all studies included.

Looking at food-based therapies, such as a ‘low red and processed meat diet’ [57] and ‘low-refined, low-carbohydrate diets’ [58] did not show any benefits, whereas only 16% of patients sticking to a semi-vegetarian (reduced meat consumption) diet relapsed after two years, compared to 67% of the omnivorous diet group [59]. Levine et al. [60] performed a CD exclusion diet (CDED), which suggests the exclusion of foods thought to increase intestinal permeability and therefore inducing colonic inflammation and dysbiosis. They performed a randomized prospective trial over 12 weeks with children suffering from mild or moderate CD: the participants were allocated to a group receiving CDED and 50% of calories from enteral nutrition (EN) for 6 weeks, followed by CDED with 25% partial EN for further 6 weeks or a group, which was on EEN for 6 weeks, followed by a free diet with 25% of EN for 6 more weeks. After 12 weeks, 75.6% of children receiving CDED and EN were in steroid-free remission compared to 45% of participants given EEN and, afterwards, partial EN. In addition, the compound of the gut microbiome was assessed, and it showed a similar pattern in both groups during the induction of the remission, but there was a rebound in the abundance of proteobacteria in the control group during the maintenance phase, compared to a sustained reduction in the CDED/EN group. The better results of the CDED/EN group might be explained by the fact, that these children displayed a better compliance to this diet. Further dietetic approaches with a positive result in reducing the risk of a relapse focus on a high ratio of ω-6/ω-3 fatty acids [61], fibre intake [61], and overall processed food [62].

## 6. Probiotics

Similar to nutritional treatments, the aim of using probiotics is to modify the gastrointestinal microbiome and the immune response, and thereby impact positively on the course of IBD. Organisms mostly present in probiotic preparations include *S boulardii*, *Lactobacillus GG*, *EcN 1917*, and *VSL #3*. A meta-analysis of over a thousand adult IBD-patients [63] showed a higher relapse rate with placebo than VSL #3; whereas S boulardii and Lactobacillus GG showed no advantage compared to placebo, EcN 1917 had a similar efficacy compared to mesalazine. In the only randomized controlled paediatric trial [64], Lactobacillus GG or a placebo was given additionally to standard maintenance therapy, and no significance was observed between the two groups. Based on absent evidence for objective benefit, current guidelines advise not to use probiotics to induce or maintain remission in patients with CD [20]. There is one study of children with UC where EcN1917 was found to be as effective as mesalamine in maintaining remission [65], and a systematic Cochrane review has highlighted methodological limitations [66]; therefore, recommendations for the treatment of patients with UC in childhood mention that probiotic agents (VSL #3, Escherichia coli Nissle 1917) may be considered an adjuvant therapy [19].

## 7. Stool Transplantation

Another approach to correct a dysbiosis is to perform a faecal microbiota transplantation. A systematic review of [67] four randomized trials with a total of 277 patients suffering from UC showed that 8 weeks after the intervention, 37% of participants were in remission, compared to 18% of the control group. Overall, 7% (10/140) of patients received a stool transplantation, and 5% (7/137) of the control group experienced a serious adverse event, which included infections (clostridium difficile and cytomegalovirus), small bowel perforation, pneumonia, and the need of IV steroids or surgery. Although European guidelines of management and treatment of UC (European Crohn’s and Colitis Organisation (ECCO), [68]) mention the possibility of stool transplantation to achieve remission in adult UC patients, it highlights the need for future trials to identify the best administration route and donor type. It is not listed as an alternative treatment in the equivalent consensus on managing adult patients with CD [69]. Currently, the only clearly recommended indication to perform a stool transplantation in children is a recurrent infection with clostridium difficile [70]: the Position Paper of the North American and the European Society for Paediatric Gastroenterology recommends serum (Hepatitis A/B/C and HIV) and stool (C. diff toxin B, culture for enteric pathogens) testing for the stool donor.

## 8. Role of Surgery in IBD

CD complications may appear that require surgery in children, and it is known that in these children, disease recurrence is common after surgery and can lead to subsequent surgical intervention. Usually, endoscopic recurrence precedes clinical recurrence and is a better predictor of the risk for future surgery. Prophylactic treatment to prevent recurrence rather than treating after the disease recurs appears to be more effective in preventing further surgery. Early postoperative surveillance for disease recurrence is mandatory to prevent complications that may lead to further surgery [71].

In the presence of active inflammation in CD, surgery, whether for perianal abscess or fistula, should be limited, if possible, to draining abscesses and relieving acute symptoms. Once the inflammation has resolved, surgery may be planned electively to deal with strictures and chronic fistula [72]. Operative intervention is indicated for an abscess that is refractory to a combination of percutaneous drainage and antimicrobial and CD therapy. Abdominal surgery, even in children, is nowadays performed by laparoscopic techniques, with the advantages of faster postoperative recovery, decreased risk of wound-related complications, less formation of intraabdominal adhesions, and better cosmesis in paediatric patients [73,74]. In cases of acute severe exacerbations of ulcerative colitis (ASC), subtotal colectomy and ileostomy may be recommended, and subsequently, pouch formation may be preferred [75].

## 9. Conclusions

The incidence of IBD in children is increasing; therefore, treatment strategies and pathophysiology have been of great research interest in the last decade. Combined with the acquired insight in the gastrointestinal microbiome, diets, not only for the induction but also for the remission of the disease in children with CD, were established, whereas faecal transplantation and probiotics are still under investigation. The greatest game-changer in paediatric gastroenterology in the last decade was the approval for treatments with anti-TNF-α drugs, which are currently the first-line biological agent for the treatment of paediatric moderate/severe IBD. Measuring drug levels and drug-antibodies supports the clinician in guiding therapy strategies. In the case of not achieving the desired clinical result, immunomodulators can be added, as drug levels might increase, and the likelihood of developing drug-antibodies is smaller. Before starting this treatment, it is recommended to determine the TPMT phenotype to identify patients at higher risk of early and severe myelosuppression. Measuring metabolites is helpful for the clinician to decide whether the patient is non-compliant or failing the treatment.

A third of the patients with a paediatric onset of IBD will lose response to anti-TNF-α, so treatment needs to be escalated to Ustekinumab, Vedolizumab, or JAK inhibitors. As those are relatively new drugs, only a few observational studies in children/adolescents with IBD have been performed so far, so long-term data are not available. It is important to notice that most recommendations for the treatment of IBD in children rely on adult data, as there is a delay before paediatric trials are started with new drugs. Randomized controlled trials with drug-naïve adolescents/children are needed to assess the efficacy in the different sub-types of IBD.

## Figures and Tables

**Figure 1 children-09-00617-f001:**
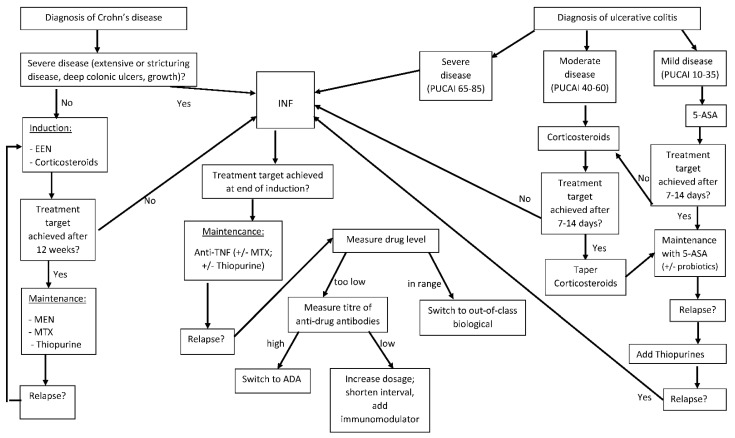
Treatment in paediatric IBD. IFX: infliximab, ADA: adalimumab, EEN: exclusive enteral nutrition, PUCAI: paediatric ulcerative colitis activity index. Adapted from ECCO/ESPGHAN Guidelines for management of paediatric IBD.

**Table 1 children-09-00617-t001:** Characteristics of monogenic Inflammatory Bowel Disease (IBD).

Young age at onset of symptoms	▪ neonatal IBD (onset within 28 days after birth)
	▪ infantile onset of IBD (IOIBD, symptoms at the age of less than 2 years)
	▪ very early-onset IBD (VEOIBD, symptoms at the age of less than 6 years)
	▪ early-onset IBD (EOIBD, symptoms at the age of less than 10 years)
Risk factors for monogenic IBD	▪ consanguinity
	▪ first-degree family members with EOIBD
	▪ first-degree family members with suspected monogenic disorder
Occurrence of primary immunodeficiencies/remarkable immune system	▪ recurrent infections in treatment-naive patients
	▪ haemophagocytic lymphohistiocytosis (HLH)
	▪ dysregulation of the immune system (e.g., IPEX or IPEX-like)
	▪ hypergammaglobulinaemia
Development of tumours	▪ B cell lymphoma
	▪ adenocarcinoma in the stomach
Different treatment options might be needed	▪ haematopoietic stem cell transplantations
	▪ surgery, parenteral nutrition, immunoglobulin replacement therapy, etc.

Adapted from the position paper from the IBD Porto Group of ESPGHAN (J. Pediatr. Gastroenterol. Nutr. 2021).

**Table 2 children-09-00617-t002:** Overview of drug monitoring.

6-TGN (pmol/8.10 RBC)	6-MMP (pmol/8.10 RBC)	Interpretation
too low (<230)	normal (<5700)	dose is too low or absent compliance
Too low (<230)	too high (>5700)	TPMT hypermetabolizer: reduce drug dose or change medication
therapeutic (230–450)	normal	treatment failure. If clinically resistant change treatment
too high (>450)	normal	low TPMT activity, reduce dose

**Table 3 children-09-00617-t003:** Monitoring anti-TNF treatment.

	Low Levels	Adequate Levels	Negative Antibody Titre
Infliximab	<5 μg/mL	>5 μg/mL	<9 μg/m
Adalimumab	<8 μg/mL	>8 μg/mL	<4 μg/m

Adapted from ECCO-ESPGHAN Guideline Update, Journal of Crohn’s and Colitis, February 2021.

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
