# Peer review of "Therapy Strategies for Children Suffering from Inflammatory Bowel Disease (IBD)—A Narrative Review"

_children, 2022, doi:10.3390/children9050617_

Round 1

Reviewer 1 Report

The authors have performed a narrative review summarising the epidemiology, assessment and therapies used in paediatric IBD. The piece has some clinical value although it is difficult to know the strength of literature justifying the suggestions and also there is a lot of data from adult patients which is used to justify use in paediatrics, so further discussion of whether adult data should be used in paediatric patients should be provided in a separate discussion section before the conclusion. I have the following additional suggestions:

  • A brief paragraph describing the search strategy to identify relevant literature would be of value for readers even if this is a narrative review.

  • Page 2, section 2 “epidemiology” The point regarding the difference in monogenic IBD noted in children would be nicely illustrated potentially with a Table outlining features that make the monogenic variants distinct from more conventional adult onset IBD and show why this should be considered its own entity.

  • Page 2, Section 3 “objective assessment/ research” - The suggestion that symptoms are sufficient to evaluated response to therapy in paediatrics rather than objective measures of disease activity is interesting and differs from adults where this is considered the standard in treat to target approaches. Please elaborate on the evidence supporting an approach of PCDAI as the treatment target rather than endoscopic activity.

  • Page 3, section 5 “corticosteroids” – it is suggested that these only be used for induction. If this is the case please provide suggestions regarding timing and regimen for weaning therapy.

  • Page 4, Section 9, immunomodulators – please provide information on hepatosplenic T cell lymphoma in paediatric patients on thiopurines, whether the effects are dependent on time on therapy and the value of EBV testing prior to commencing treatment.

Minor points:

- Biologic therapies like adalimumab and infliximab do not require uppercase initial letters. Also please keep abbreviations for infliximab and adalimumab consistent throughout text (sometimes INF and IFX are used for infliximab)

- Page 4, line 172 “Cytomegalievirus” is misspelt, please correct

- Table 2 – provide references for cited target ranges

Author Response

Dear Reviewer,

Thank you so much for your time and effort for reviewing our manuscript ‘Therapy strategies for children suffering from inflammatory bowel disease (IBD) - a narrative review’.

We have adressed all of yur comments as following:

The authors have performed a narrative review summarising the epidemiology, assessment and therapies used in paediatric IBD. The piece has some clinical value although it is difficult to know the strength of literature justifying the suggestions and also there is a lot of data from adult patients which is used to justify use in paediatrics, so further discussion of whether adult data should be used in paediatric patients should be provided in a separate discussion section before the conclusion. We have added this issue’s discussion tot he ‘objective/research’ section and also mentioned it again in the conclusion, to point out this important fact.

I have the following additional suggestions:

  • A brief paragraph describing the search strategy to identify relevant literature would be of value for readers even if this is a narrative review. Thank you very much for this important hint- we’ve added this in the section ‘objective assessment/research’.

  • Page 2, section 2 “epidemiology” The point regarding the difference in monogenic IBD noted in children would be nicely illustrated potentially with a Table outlining features that make the monogenic variants distinct from more conventional adult onset IBD and show why this should be considered its own entity. A brief summary of characteristics has been added, adapted from the IBD Porto Group Consensus 2021.

  • Page 2, Section 3 “objective assessment/ research” - The suggestion that symptoms are sufficient to evaluated response to therapy in paediatrics rather than objective measures of disease activity is interesting and differs from adults where this is considered the standard in treat to target approaches. Please elaborate on the evidence supporting an approach of PCDAI as the treatment target rather than endoscopic activity. Due to the invasiveness of endoscopies, most studies rely on measuring clinical relapses/amount of steroid-courses needed, activity scores etc., which has been more elaborated and literature regarding the correlation of activity scores and endoscopic findings has been added in the section ‘objective assessment/research’.

  • Page 3, section 5 “corticosteroids” – it is suggested that these only be used for induction. If this is the case please provide suggestions regarding timing and regimen for weaning therapy. The recommended dosage and weaning has been added.

  • Page 4, Section 9, immunomodulators – please provide information on hepatosplenic T cell lymphoma in paediatric patients on thiopurines, whether the effects are dependent on time on therapy and the value of EBV testing prior to commencing treatment. Thank you very much for this proposition in ordert to round the information about thiopurines- it has been added.

Minor points:

- Biologic therapies like adalimumab and infliximab do not require uppercase initial letters. Also please keep abbreviations for infliximab and adalimumab consistent throughout text (sometimes INF and IFX are used for infliximab). Thank you, has been changed accordingly.

- Page 4, line 172 “Cytomegalievirus” is misspelt, please correct

Has been corrected accordingly, thank you.

- Table 2 – provide references for cited target ranges

This has been added.

Please let us know if you have any questions or other suggestions,

Sincerely,

  1. Légeret

Reviewer 2 Report

Overall, this narrative review aims to provide an overview of the current therapeutic options for management of pediatric IBD (including CD, UC, and IC), and also offers an evidence-based, step-wise approach to management of  IBD in the clinical setting. 

As a general perspective, I am concerned that while this review is well-researched, it does not necessarily add a significant amount to the literature base as there are already numerous practical guides to pediatric IBD management as put forth through NASPGHAN/ESPGHAN and through other forums including a very similarly outlined review by Guariso G and Gasparetto M in the World Journal of Gastroenterology (2017).

In addition, I feel that though this manuscript is well-outlined, the writing is overall muddled and lacks good transitions to make the arc of the piece easily digestible by readers. 

With regard to specific section-based comments: 

3. Objective Assessment / Research 

Most narrative reviews define the mechanisms through which research is conducted (i.e. which data bases, key words, etc.) to highlight the rigor of the review. It is not clear to me how all of the data was collected for this report. 

I would also recommend including a conclusion statement at the end of this section that helps readers anticipate that the remainder of the sections will specifically be centered on treatment overview/approach to management. There is no transition at this time, so the start of the next section feels very abrupt. 

4. 5-ASA

While I agree that there are only smaller existing studies that highlight the efficacy of 5-ASA use in pediatric IBD, I would recommended including some of these citations such as those by Zeisler et al in JPGN (2013) and Sokolik et al in the Journal of Clinical Gastroenterology (2018); both of these studies included 200 patients and demonstrated positive outcomes.

5. Corticosteroids

The authors state that EEN is preferred to corticosteroids to induce remission in CD (reference 16, line 105). However, a recent meta-analysis by Yu et al in the World Journal of Pediatrics (2019) demonstrated no significant difference in remission rates between EEN and corticosteroid groups. 

7. Probiotics 

8. Stool transplantation

Given that there is very little efficacy data with these therapies, I do wonder if the authors might consider moving these sections to the very end of the manuscript (i.e. after Anti-TNF alpha therapy).

9. Immunomodulators 

Would ensure that this section begins with an overview statement on which drugs will be discussed. Additionally, I believe that table 1 versus table 2 should follow rather than precede line 211, which is currently not the case.

I also believe that table 2 should not be in this section, but rather in the next section on Anti-TNF alpha therapy (after line 246). 

10. Anti-TNF alpha therapy

Would recommend discussing use of IFX OR ADA upfront to induce remission in more severe IBD presentations. As the manuscript later discusses in this section, there are no studies that necessarily distinguish between using one versus the other biologic upfront — though it is certainly true that given the earlier FDA approval of IFX there is more collective data on use of this agent. 

Line 267 provides a very helpful practical guide to IBD management. However, I am worried that there is no clear citations/credit given to figures in other publications that are very similar (including on UpToDate and in the NASPGHAN and ESPGHAN guidelines). 

Author Response

Dear Reviewer,

Thank you so much for your time and effort for reviewing our manuscript ‘Therapy strategies for children suffering from inflammatory bowel disease (IBD) - a narrative review’.

We have adressed all of yur comments as following:

Overall, this narrative review aims to provide an overview of the current therapeutic options for management of pediatric IBD (including CD, UC, and IC), and also offers an evidence-based, step-wise approach to management of  IBD in the clinical setting. 

As a general perspective, I am concerned that while this review is well-researched, it does not necessarily add a significant amount to the literature base as there are already numerous practical guides to pediatric IBD management as put forth through NASPGHAN/ESPGHAN and through other forums including a very similarly outlined review by Guariso G and Gasparetto M in the World Journal of Gastroenterology (2017).

In addition, I feel that though this manuscript is well-outlined, the writing is overall muddled and lacks good transitions to make the arc of the piece easily digestible by readers. 

With regard to specific section-based comments: 

  1. Objective Assessment / Research

Most narrative reviews define the mechanisms through which research is conducted (i.e. which data bases, key words, etc.) to highlight the rigor of the review. It is not clear to me how all of the data was collected for this report. Thank you for this important hint- it has been pointed out by the other reviewer as well, and has been changed accordingly

I would also recommend including a conclusion statement at the end of this section that helps readers anticipate that the remainder of the sections will specifically be centered on treatment overview/approach to management. There is no transition at this time, so the start of the next section feels very abrupt. This section has been changed accordingly.

  1. 5-ASA

While I agree that there are only smaller existing studies that highlight the efficacy of 5-ASA use in pediatric IBD, I would recommended including some of these citations such as those by Zeisler et al in JPGN (2013) and Sokolik et al in the Journal of Clinical Gastroenterology (2018); both of these studies

included 200 patients and demonstrated positive outcomes. Those two papers have been added as references.

  1. Corticosteroids

The authors state that EEN is preferred to corticosteroids to induce remission in CD (reference 16, line 105). However, a recent meta-analysis by Yu et al in the World Journal of Pediatrics (2019) demonstrated no significant difference in remission rates between EEN and corticosteroid groups. We’ve added this meta-analysis and tried to make the benefit of EEN more understandable (less side-effects).

  1. Probiotics
  2. Stool transplantation

Given that there is very little efficacy data with these therapies, I do wonder if the authors might consider moving these sections to the very end of the manuscript (i.e. after Anti-TNF alpha therapy). As the outline/structur was not clear to reviewer one, it was written more clearly that the manuscript follows a ‘step-up’ therapy regime, therefore we have to leave the order, we are afraid.

  1. Immunomodulators

Would ensure that this section begins with an overview statement on which drugs will be discussed. Additionally, I believe that table 1 versus table 2 should follow rather than precede line 211, which is currently not the case. I also believe that table 2 should not be in this section, but rather in the next section on Anti-TNF alpha therapy (after line 246).

We’ve added an introduction sentence and changed table 1 (now being table 2) to the section ‘anti-TNF alpha’ according to your suggestion.

  1. Anti-TNF alpha therapy

Would recommend discussing use of IFX OR ADA upfront to induce remission in more severe IBD presentations. As the manuscript later discusses in this section, there are no studies that necessarily distinguish between using one versus the other biologic upfront — though it is certainly true that given the earlier FDA approval of IFX there is more collective data on use of this agent. According studies -discussion IFX as firs-line treatment to induce and maintain remission – have been added.

Line 267 provides a very helpful practical guide to IBD management. However, I am worried that there is no clear citations/credit given to figures in other publications that are very similar (including on UpToDate and in the NASPGHAN and ESPGHAN guidelines). 

Has been added accordingly, thank you for the hint.

Please let us know if you have any questions or other suggestions,

Sincerely,

  1. Légeret

Round 2

Reviewer 1 Report

The authors have adequately addressed all the suggested points and have completed an interesting narrative review. I would advise checking the grammar of the piece thoroughly.

Author Response

Comments Reviewer 1:

  • The authors have adequately addressed all the suggested points and have completed an interesting narrative review. I would advise checking the grammar of the piece thoroughly.

Thank you, grammar was checked by a native english speaking person

Reviewer 2 Report

Did a nice job of responding to concerns. I do think that some of the transitions still feel abrupt, but improved markedly.

Author Response

Comments Reviewer 2:

  • Did a nice job of responding to concerns. I do think that some of the transitions still feel abrupt, but improved markedly.

Thank you, we have included new headings that cover the different treatment options and now transitions should not feel abrupt anymore
